# Analysis of cortical dysplasias using b-tensor encoding diffusion MRI in an animal model

Olimpia Ortega-Fimbres[1‡], Ricardo Ríos-Carrillo[2‡], Edith Gaspar-Martínez[1],
Priscila Ruiz-Acosta[1], Mirelta Regalado[1], Hiram Luna-Munguía[1],
Alonso Ramírez-Manzanares[3], Luis Concha[1]*

1 Instituto de Neurobiología, Universidad Nacional Autónoma de México, Querétaro, México, 2 Centre for Functional and Metabolic Mapping, Robers Research Institute, Western University, London, Ontario, Canada, 3 Departamento de Ciencias de Computación, Centro de Investigación en Matemáticas, A.C., Guanajuato, México

‡ These authors contributed equally to this work.
* lconcha@unam.mx

## Abstract

Cortical dysplasias are malformations of cortical development characterized by disorganization of the cyto- and myeloarchitecture of the neocortex. They are a common cause of epilepsy and their diagnosis through conventional imaging can often be challenging, hindering surgical treatments. Diffusion-weighted magnetic resonance imaging (dMRI) has the ability to infer tissue properties at the microscopic scale, making it a promising technique for detection of cortical dysplasias. This study aims to assess the microarchitecture of the cerebral cortex in a murine model of cortical dysplasia using dMRI acquired with b-tensor encoding. Pregnant Sprague-Dawley rats were administered either carmustine (BCNU) or saline solution on day 15 of gestation. Their offspring were imaged at 120 days of age using a 7 tesla scanner, acquiring diffusion-sensitive images with b-tensor encoding. Images were processed with Q-space trajectory imaging with positivity constraints (QTI+) to derive various metrics along a curvilinear coordinate system across the neocortex. After scanning, the brains were processed for immunofluorescence and histological examinations. Experimental animals exhibited a significant reduction of microscopic fractional anisotropy (µFA) and anisotropic kurtosis ($K_{shear}$) in the middle and lateral cortical layers compared to the control animals. Immunofluorescence and histological analysis showed decreased and dysorganized myelinated fibers, and an increase of glial processes in BCNU-treated animals. Given the applicability of b-tensor encoding in clinical scanners, this approach holds promise for improving detection of focal cortical dysplasias in patients with epilepsy.

**Data availability statement:** The raw diffusion-weighted images and photomicrographs are available at https://osf.io/n46d8 (https://doi.org/10.17605/OSF.IO/N46D8).

**Funding:** Funding for this project was provided by Conahcyt/Secihti (CF-2023-I-218 LC) and UNAM-DGAPA (IN204720 and IN213423 LC, IN211326 HLM). The first author received a scholarship from SECIHTI (1019538). The funders had no role in study design, data collection and analysis, decision to publish, or preparation of the manuscript.

**Competing interests:** The authors have declared that no competing interests exist.

# 1 Introduction

The cerebral cortex exhibits a highly organized architecture. Alterations during prenatal neurodevelopment can lead to diverse anatomical abnormalities that vary in extent and severity as a consequence of the precipitating insult. Focal cortical dysplasias (FCDs) are a specific type of malformation of cortical development, representing the first and second cause of pharmacoresistant focal epilepsy in children and adults, respectively [1]. These malformations are characterized by disrupted cortical layers, neuronal heterotopia, and presence of dysmorphic neurons [2]. Magnetic resonance imaging (MRI) is the primary diagnostic tool used for detection of FCDs and to guide surgical interventions, when appropriate [3]. Diagnosis of FCDs can be challenging, as their detection relies on subtle visual cues such as focal cortical thickening, slight hyperintensity, and blurring of the gray/white matter boundary on conventional T1- or T2-weighted MRI [4]. Moreover, there can be ample variation in their size and anatomical location [5–7]. This results in frequent underdiagnosis and inadequate treatment, highlighting the need for more sensitive and specific imaging methods [4]. Detection of FCDs can be improved substantially through quantitative methods that augment the diagnostic yield of conventional MRI, such as texture analysis [8], voxel-based morphometry [9], and artificial intelligence [10,11], and more recently through acquisition and analysis of MRI fingerprinting [12,13].

Given that anatomical abnormalities that usually accompany FCDs can be subtle, there is a need for imaging methods that are able to capture the histopathological features that characterize these lesions. Diffusion-weighted MRI (dMRI) offers an alternative, non-invasive approach for studying tissue microarchitecture by measuring the diffusion of water molecules in different tissues [14–16]. This technique can capture structural details that are not visible with conventional MRI, making it potentially more effective for diagnosing FCD [17]. Furthermore, dMRI can be applied to both human and animal models, enabling the translation of experimental findings to clinical practice [18,19]. Several methods to analyze the diffusion signal have been introduced, the majority of which are implemented on data acquired using single diffusion encoding [20]. While these methods have been mainly used to characterize white matter, they can also provide relevant information regarding the laminar and columnar structure of the human, non-human primate, and rodent neocortex [21–27]. Recent advances in dMRI acquisition and analysis methods have further improved the diagnostic capacity of this technique by providing more information on tissue characteristics [28,29]. Multidimensional encoding techniques have been developed to allow a more thorough analysis of brain microstructure by encoding diffusion through complex, time-varying gradient waveforms [30,31]. In conventional single diffusion encoding, diffusion weighting is applied using gradients of various orientations and magnitudes (b-values), and the encoding can be fully described by a single vector. In multidimensional encoding, this concept is generalized: the diffusion-encoding vector is replaced by a second-order tensor (the b-tensor), allowing control not only over orientation and magnitude, but also over the shape of the diffusion encoding. Different b-tensor shapes (e.g., linear, planar, or spherical) query the tissue in complementary ways and yield information not reachable by single diffusion encoding.

Indeed, while diffusion tensor imaging (DTI) provides characteristics of a single diffusion tensor per voxel, b-tensor encoding adds a metric of the covariance of domain D-tensors, which allows sampling properties of intra-voxel diffusion tensor distribution and provides rich information that better accounts for the heterogeneity of nervous tissue components [32]. These innovations make b-tensor encoded dMRI a promising tool for characterizing the microarchitecture of the cortex and alterations present in FCDs and other cortical malformations [33].

In this study, we used advanced dMRI techniques to characterize tissue properties of the cortex in an animal model of FCD. Histological analyses were performed to bridge tissue properties with water diffusion patterns in the regions of interest. The aim was to assess whether b-tensor encoded dMRI methods are sensitive to the histopathological features of FCD as an initial exploration of their clinical applicability.

## 2 Methods

All experimental procedures were approved by the Ethics Committee of the Institute of Neurobiology (Universidad Nacional Autónoma de México; protocol 111-A).

### 2.1 Murine model of cortical dysplasia

To induce the histopathological features of FCD Type IIa in rodents, we used a known animal model that disrupts corticogenesis *in utero*. Sprague-Dawley rats were injected with a single dose of either the alkylating agent BCNU (bis-chloroethylnitrosourea, also known as carmustine; 20 mg/kg in saline solution, i.p., 5; 5 rats), or saline solution (for control; 3 rats) on embryonic day 15; a time point that corresponds to the peak of cortical neurogenesis [34]. Based on previous reports [35,36] and our observations [27,37], this procedure is known to induce cortical alterations in the pups similar to those found clinically [1,38]. The pups were kept with their mothers until weaning. All animals had access to food and water *ad libitum* and were always kept at the animal facility under controlled environmental conditions. Experiments were only conducted with the offspring. A total of 18 control (6 female) and 20 BCNU-treated (8 female) rats were included for further analysis. None of the animals died, nor displayed any signs of poor quality of life, suffering, or distress throughout the study.

### 2.2 Diffusion-weighted magnetic resonance imaging

Images were acquired at the National Laboratory for Magnetic Resonance Imaging (Lanirem) in Juriquilla, Queretaro, Mexico, using a 7 T Bruker Pharmascan preclinical MRI scanner and a 2×2 array head surface coil. The rats were anesthetized with isoflurane (4% for induction, 2% for maintenance) and kept warm by circulating water at 37 °C through hoses placed under the scanner bed. Vital signs were continuously monitored throughout the study using a compatible system. A single imaging session was performed for each animal (four months old) to determine diffusion parameters in the dysplastic and normal cortices. Each session lasted one hour. dMRI were obtained using an open-source sequence based on a 2D spin-echo echo-planar acquisition sequence, available from the Preclinical Neuro MRI repository (https://github.com/mdbudde/mcw_Preclinical_MRIsequences). Images with coronal orientation were acquired with voxel resolution of 200×200×1010 μm$^3$, repetition time (TR) = 2000 ms, echo time (TE) = 40.86 ms, flip angle = 90°. The implemented protocol consisted of three b-tensor shapes: linear, spherical and planar [39]. The spherical tensor encoding (STE) gradients were numerically optimized and compensated for concomitant gradients prior to acquisition by using the NOW toolbox (https://github.com/jsjol/NOW) [40,41] tailored to minimize TE and scaled in magnitude to obtain four b-values (200, 700, 1400, and 2000 s/mm$^2$). To retain gradient spectral characteristics between waveforms, the planar and linear tensor encoding gradients (LTE and PTE, respectively) were extracted from the STE waveform, using one axis for LTE and the other two for PTE [30]. The STE waveform was rotated in 10 directions at each b-value; LTE and PTE waveforms were rotated and scaled to obtain [10,16,46] directions for each corresponding b-value. S1 Fig shows the waveforms and protocol used in this study. The raw DWI are available at https://osf.io/n46d8.

## 2.3 DWI processing

The images were preprocessed to minimize noise and artifacts. This included noise reduction [42] and correction of geometric inhomogeneities [43] using Mrtrix 3.0.4 [44] and fsl 6.0.7.1 [45] (Fig 1A). Q-space trajectory imaging with positivity constraints (QTI+) [46] was computed as implemented in https://github.com/DenebBoito/qtiplus. In addition to the four diffusion-tensor metrics, namely fractional anisotropy (FA), axial, radial and mean diffusivities (AD, RD and MD, respectively) (Fig 1B, top panel), QTI+ provides four complementary metrics: microscopic anisotropy (µFA), microscopic orientation coherence (CC), and mean anisotropic and isotropic kurtosis ($K_{shear}$, and $K_{bulk}$, respectively) [29] (Fig 1B, bottom panel). µFA quantifies the average anisotropy of compartments in a voxel disregarding their orientation, while CC is sensitive to the intra-voxel variability of their orientations. $K_{bulk}$ informs of the heterogeneity in isotropic diffusivities (i.e., tensor sizes) across compartments, and is related to the variance of cell density, edema, and free water. $K_{shear}$ informs of the variance in microscopic anisotropy and dispersion of microenvironments (i.e., tensor shapes and orientations), and helps separate tissue components by their geometry (e.g., axonal from non-axonal components). There is discrepancy in the literature related to the notation of some diffusion metrics [33]. In this work we use the notation described in [29]. These summary metrics simplify the interpretation of the distribution of tensors of different shapes, sizes, and orientations, and

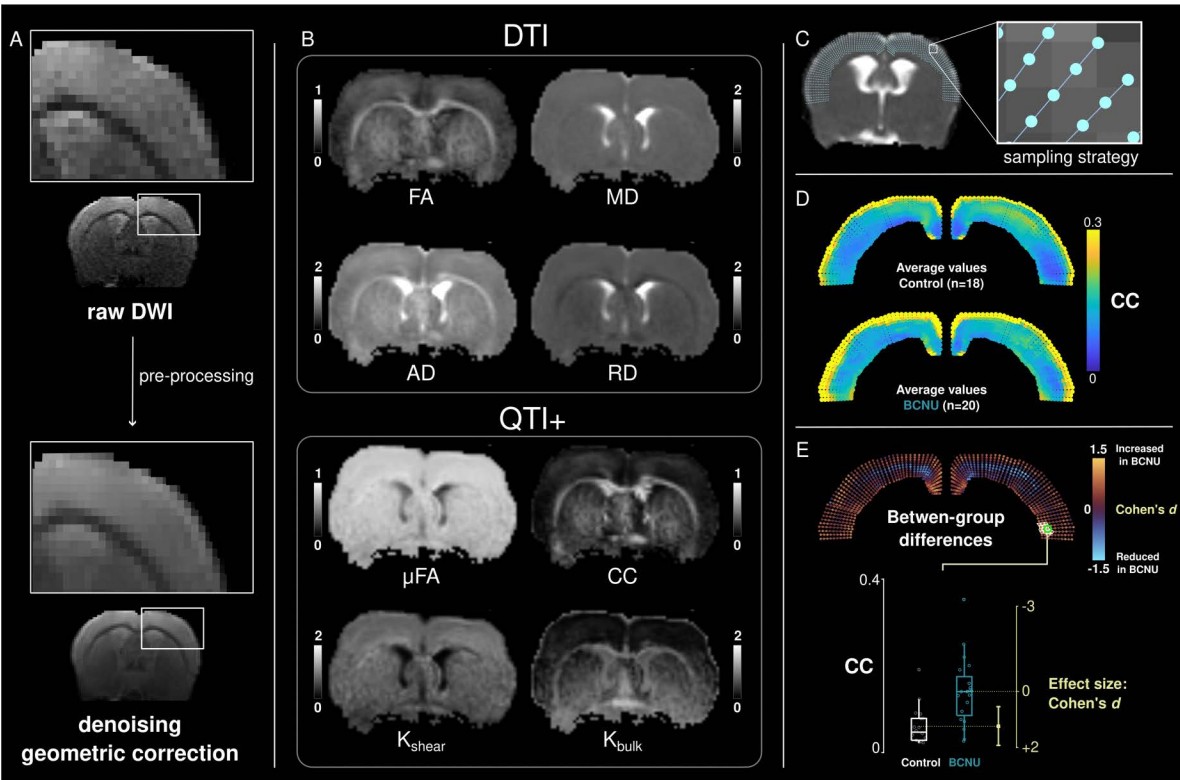

**Fig 1. Analysis pipeline.** A: Raw dMRI were preprocessed to minimize noise and correct for geometric inhomogeneities. B: QTI+ provided four DTI metrics (top): fractional anisotropy (FA), and axial, radial and mean diffusivities (AD, RD, MD, respectively; in units of mm²/s), as well as (bottom) microscopic anisotropy (µFA), microscopic orientation coherence (CC), and anisotropic and isotropic kurtosis ($K_{shear}$ and $K_{bulk}$, respectively). C: A curvilinear coordinate system (50 grid-lines spanning medial to lateral, and 10 depth levels from the pial border to the gray/white matter border) was defined in each rat brain, providing inter-subject anatomical correspondence. Descriptive statistics were obtained from the data sampled by the grid-lines, the point-wise average values are shown for each group (D). Student's *t*-tests were subsequently performed at each point followed by correction for multiple comparisons (E). Marker size and color indicate effect size (Cohen's *d*), gray circles indicate point-wise $p_{uncorr}$ <0.01, white areas indicate significant clusters ($p_{clus}$ <0.05). A Gardner-Altman plot presents data from one exemplary vertex (green circle in Cohen's *d* map).

are sensitive to features of microarchitecture [28]. The valid range for FA, μFA and CC was between 0 and 1, and voxels outside of these ranges (less than 0.5% of all voxels) were attributed to fitting errors and excluded from further analyses.

## 2.4 Spatial analysis

A curvilinear coordinate system with anatomical correspondence between animals was constructed for each rat to sample dMRI parameters across the depth and extent of the cortex [27]. For this purpose, in a single coronal slice (located approximately −0.8 mm posterior to bregma, harboring primary motor and somatosensory cortices [M1 and S1, respectively]), the pial surface of the brain and the boundary between gray and white matter were manually delineated. A Laplacian potential field was simulated between these two boundaries [47]. From the pial surface, 50 evenly distributed virtual trajectories (grid-lines) were created and propagated organically through the Laplacian field; each trajectory extended towards the white matter boundary, following the curvature of the cortex in a manner analogous to cortical columns. The dMRI parameters were sampled at 10 equidistant points along each of these grid-lines (Fig 1C). The code to create this curvilinear grid is available at https://github.com/lconcha/Displasias.

## 2.5 Statistical analysis

Acknowledging that cortical cyto-and myeloarchitecture varies among cortical regions [48], the analysis was conducted in a spatially dependent fashion. Abnormalities were assessed separately for each dMRI metric using univariate statistics (Fig 1D, E). At each point in each grid-line, Student's $t$-tests were performed to detect differences between the two groups (Fig 1E). Effect size estimation (the magnitude of the difference between groups) was calculated using Cohen's $d$. Statistical analyses were corrected to minimize the probability of Type I errors (false positives) by using cluster-level permutation tests [49]. The vertex-wise cluster-forming threshold was set as $p_{uncorr} < 0.01$. Cluster significance ($p_{clus}$) was determined by randomizing the data between experimental groups, performing 5,000 permutations to obtain the distribution of cluster sizes that could arise by chance. From this empirically-derived null distribution, the probability of finding clusters with a similar extent to those observed in the real data (i.e., without randomization between groups) was calculated. Cluster-wise statistical significance was defined as $p_{clus} < 0.05$.

## 3 Histological analysis

After completing the dMRI studies all the animals were deeply anesthetized using an intraperitoneal overdose of sodium pentobarbital and intracardially perfused with 0.9% NaCl solution followed by 4% paraformaldehyde (PFA) solution. The brains were removed and preserved in fresh PFA 4% solution for 24 h at 4 °C. After this, each brain was immersed in a 20% sucrose solution for 48 h, followed by a 30% sucrose solution for another 48 h. Brains were stored at −72 °C until further analysis. Coronal sections (20 μm-thick) from the region of interest were obtained using a cryostat (Leica) based on the following Paxinos and Watson Rat Brain Atlas interaural coordinates: 8.74–8.08 mm. Slices were kept in a cold 1X phosphate buffer solution (PBS; Sigma-Aldrich). Immunofluorescence was performed using the primary antibodies anti-Myelin Basic Protein (MBP; 1:500; abcam), anti-Neuronal Nuclear Protein (NeuN; 1:350; abcam), and anti-Glial Fibrillary Acidic Protein (GFAP; 1:350; Sigma-Aldrich). For the triple immunofluorescence staining, the tissue sections were blocked with Bovine Serum Albumin (BSA; Sigma-Aldrich) 2% solution + 0.3% triton X-100 (ThermoFisher) in 1X PBS for 45 min. The sections were incubated with the primary antibodies (MBP and NeuN) for 24 h at 4°C. Then, slices were washed five times for 10 min in PBS 1X + Tween 0.1% solution (Sigma-Aldrich). Secondary antibodies conjugated with fluorescent dyes (AlexaFluor, goat anti-mouse-647 and goat anti-rabbit-555) were diluted 1:500 in a solution of PBS 1X + 0.1% Tween and incubated for 4 h at 4°C. After incubation, the slices underwent five washes (each one lasting 10 min) in PBS 1X. The sections were then blocked with a solution of BSA 2% + 0.3% Triton X-100 in PBS 1X for 45 min. Finally, GFAP was added in a PBS 1X + 0.1% Tween solution and incubated for 24 h at 4°C. Afterwards, another set of five 10-min washes in PBS 1X + 0.1% Tween was done, and the corresponding secondary antibody was added (AlexaFluor, goat anti-mouse-488) for 4 h, followed by five additional washes in 1X PBS.

Finally, the slices were mounted using Mowiol. Brain slices were imaged using a confocal microscope (Zeiss LSM 880, with 488/594/647 nm wavelengths) and a fluorescence microscope (Zeiss Apotome, with 488/594 nm wavelengths). The system of this last microscope was linked to a computer running AxioVision software (version 4.8), where the MosaiX module was used to acquire mosaic images at 10X. Mosaics are available at https://osf.io/n46d8.

The photomicrographs were evaluated using Fiji [50]. Samples were taken from the primary motor cortex (M1) and the primary somatosensory cortex (S1). In these samples, the brightness threshold was automatically adjusted [51], and the spatial profile of glial density was also determined by calculating the percentage of the area occupied by the cells. The organization of the myeloarchitecture was evaluated through structure tensor analysis [52], as implemented in OrientationJ (https://github.com/Biomedical-Imaging-Group/OrientationJ) [53], calculating vector and local coherency maps using a Gaussian window of 15 μm. From these, we performed between-group comparisons of texture coherency and energy metrics, as well as principal texture orientation with respect to the pial boundary, using Student's *t*-tests and computed profiles of said metrics as a function of cortical depth.

## 4 Results

### 4.1 Analysis of diffusion-weighted magnetic resonance images

All diffusion metrics showed spatial variability across the extent and depth of the cortex, which is not captured by histogram analyses of the entire cortex (Figs 2 and 3). In control animals, the middle layers of the cortex showed the highest FA and lowest RD in the middle layers of M1 and S1 regions. Contrarily, the most lateral aspects of the cortex showed the lowest average FA values. Mean and axial diffusivities were less heterogeneous across the cortex, with the exception of the most superficial layers. Metrics derived from QTI+ in control animals showed very high values of μFA in the middle

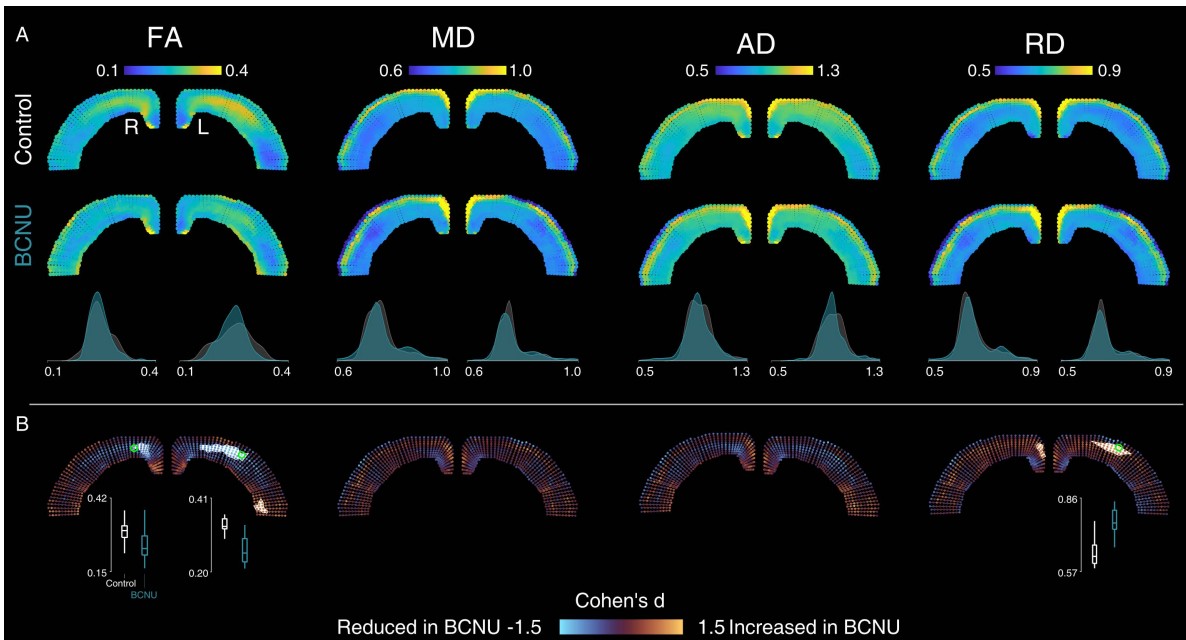

**Fig 2. DTI metrics.** A: Average values for each metric are shown for control and BCNU-treated animals (first two rows). Histograms show the group-wise average values across the entire cortex per hemisphere (third row). B: Between-group differences illustrated as in Fig 1E. Effect sizes (Cohen's *d*) are color-coded at each point per grid line. White areas indicate cluster-corrected statistical significance (p$_{clus}$ <0.05). Box plots of vertices identified in green are shown below hemispheres with significant clusters.

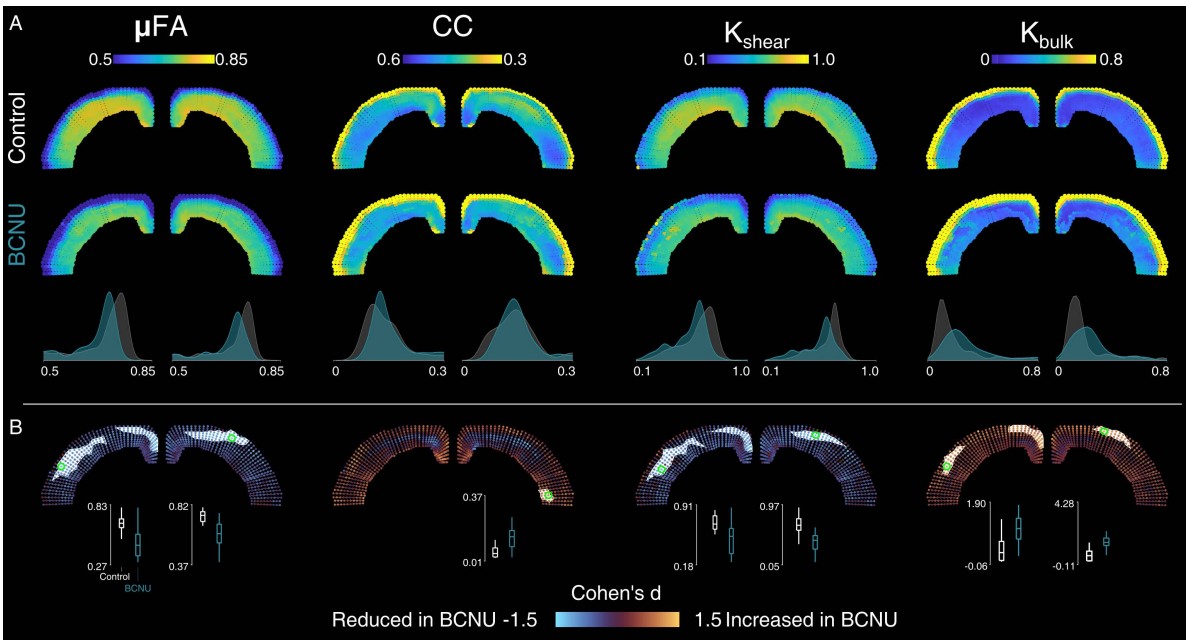

**Fig 3. QTI+ metrics.** Figure layout and description as in Fig 2.

layers of the cortex with decreasing values towards the lateral aspects of the cortex. This pattern was mirrored by $K_{shear}$. A band of very high CC and $K_{bulk}$ values was seen in the most superficial layers of the cortex. Average between-group differences are readily visible for µFA, $K_{shear}$, and $K_{bulk}$ in the spatial maps and in the histogram analyses.

BCNU-treated animals showed decreased FA values in the central-medial areas of the cortex in both hemispheres, encompassing the motor cortices M1 and M2, the primary somatosensory, and the cingulate cortex (Fig 2). This reduction of FA was accompanied by an increase of axial diffusivity and a reduction of axial diffusivity that was significant at the cluster level in the left hemisphere. Mean and axial diffusivities did not show any statistically significant differences between groups. QTI+ metrics attainable only through b-tensor encoding provided additional information (Fig 3). Both microscopic anisotropy and anisotropic kurtosis showed extended reductions along the cortex of both hemispheres in BCNU-treated animals (including cingulate, motor, and somatosensory cortex), mostly in the middle to superficial layers. The deep layers of the most lateral aspect of the left hemisphere showed an increased of CC, while the middle layers of the rest of the cortex showed reductions that were not significant at the cluster level. Finally, there were distributed increases of $K_{bulk}$ throughout the superficial layers of the cortex in both hemispheres.

## 4.2 Histology

Qualitative evaluations revealed reduced intracortical myelination in BCNU-treated rats, with reduced MBP+ fibers in the most superficial cortical layers. Additionally, variations in the distribution of neuronal nuclei (NeuN+) and morphological/quantitative changes in astrocytes were observed in BCNU-treated rats compared to the control group (Fig 4).

**4.2.1 Myelin analysis.** Quantitative examination of histology through structure tensor analysis of MBP-labelled immunofluorescent photomicrographs revealed differences of the myeloarchitecture of the cerebral cortex between the control and experimental animals. BCNU-treated animals showed decreased coherence from the medial to the most lateral region of the cortex. This was further corroborated by vector maps, where interspersed vector vortices were observed in the somatosensory region, indicating a disorganization of myelin fibers (Figs 5 and S2).

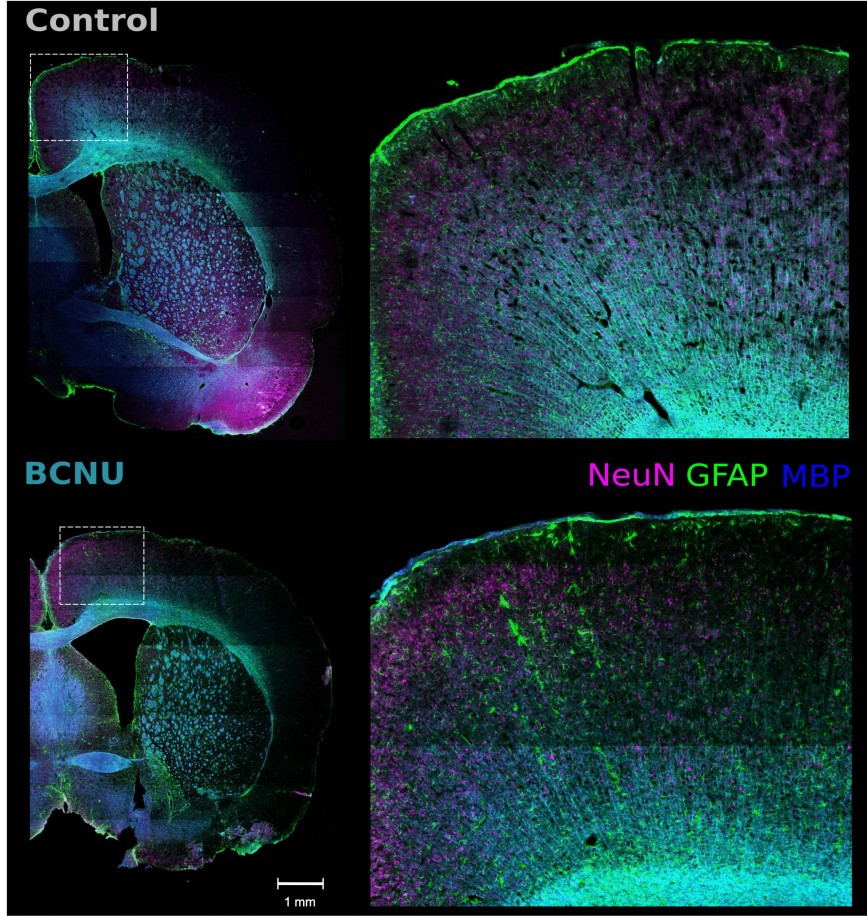

**Fig 4. Triple-labelled Immunofluorescence showing neuronal nuclei (NeuN; in magenta, 594 nm), glia (GFAP; in green, 488 nm), and myelin-ated fibers (MBP; in blue, 647 nm) from a control (top) and a BCNU-treated rat (bottom).** The experimental animal shows overall less intracortical myelin, heterogeneous spatial distribution of neuronal nuclei, and increased astrocytic processes.

**4.2.2 Histological analysis of astrocyte cortical distribution.** There was an overall increase of the percentage of area occupied by astrocytes (GFAP+) between the control and experimental groups in the primary motor cortex (Fig 6, top row). Depth-wise analysis revealed increased GFAP+ area particularly in cortical layers IV-VI (Fig 6, middle panel). This increased presence of glial processes was not as marked in the S1 region (Fig 6, bottom row).

## 5 Discussion

Despite the high spatial resolution attainable with modern anatomical MRI, identification of FCD in patients with focal-onset epilepsy remains a challenge, as their microscopic disarray can hide behind tissue that looks deceptively ordinary at macroscopic scales. In this work we show that advanced dMRI with b-tensor encoding is able to extract information at the mesoscopic level, evidencing subtle histopathological landmarks characteristic of cortical malformations.

We used an animal model that presents the subtle histological cyto- and myeloarchitectonic irregularities character-istic of human FCD [27,34,37]. In prior work from our group using the same model we showed that multi-tensor fit of the diffusion signal was able to separate groups of intracortical fibers depending on their orientation to the cortical surface (i.e., radial and tangential fibers) and, moreover, differentially identify diffusion abnormalities in BCNU-treated animals

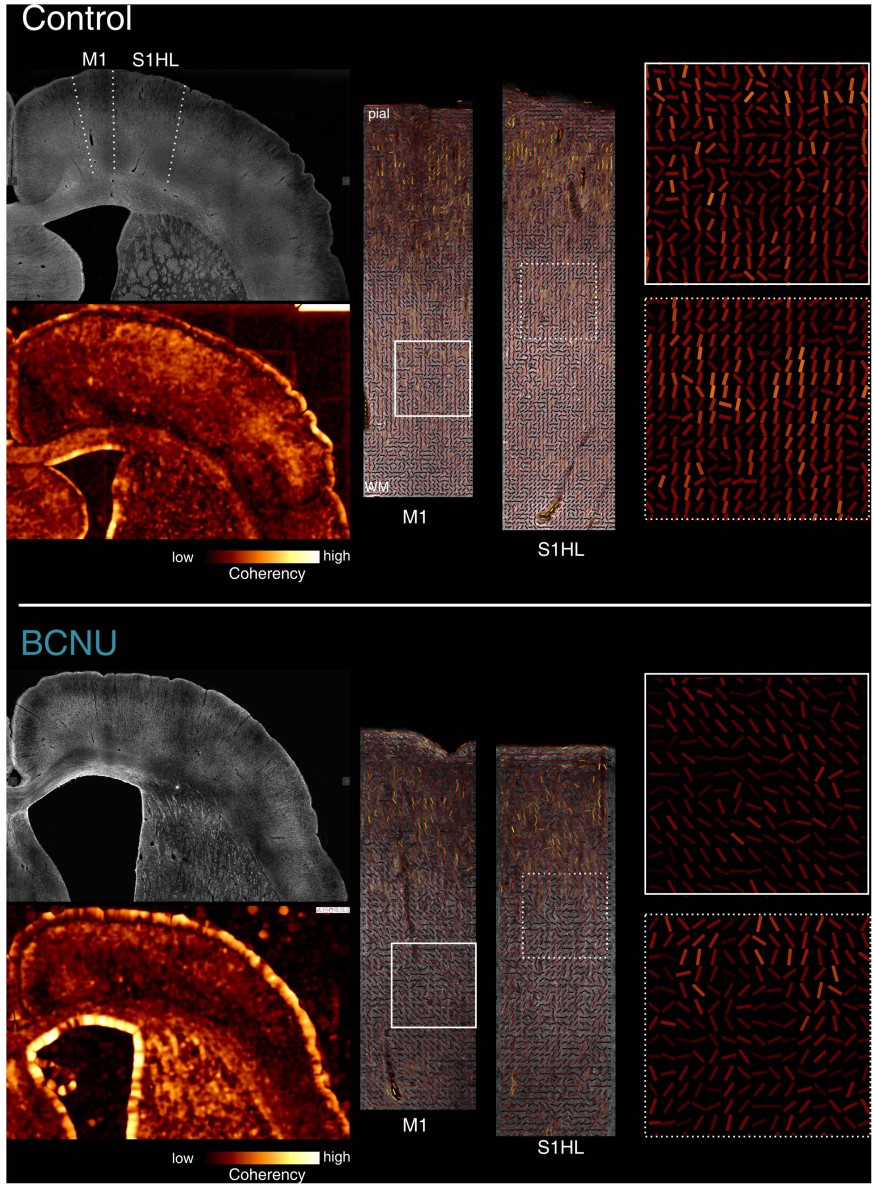

**Fig 5. Structure tensor analysis of MBP+ fibers.** Exemplary control (top) and BCNU-treated (bottom) animals showing MBP+ fibers (grayscale) and corresponding coherency maps (warm colors) derived from structure tensor analysis. For each animal, primary motor (M1) and primary somatosensory cortex (hindlimbs region, S1HL) are shown enlarged in the middle panels. Solid and dotted squares indicate the regions magnified in the rightmost panels. Experimental animals show reduced coherency throughout the cortex and magnifications illustrate coherent directions that are observed in the control group, while experimental animals show less coherency and interspersed vortices. Depth-wise analyses for M1 and S1HL are shown in S2 Fig.

[27]. Here, we used QTI+ to derive diffusion tensor metrics, and found reductions of FA in the middle cortical layers, in close agreement with our previous report [27], despite the age difference of the rats between the two studies (P30 vs P120). These results are in line with previous reports of reduced FA within FCD lesions [54,55] and also in the superficial white matter adjacent to the lesions [56–58]. DTI is a commonplace method, available in virtually all MRI scanners. The robustness of the FA findings across studies indicates that this metric has the potential to aid in the identification of FCD in patients [59]. Its sensitivity, however, is likely reduced as a consequence of the known limitations of DTI that render it

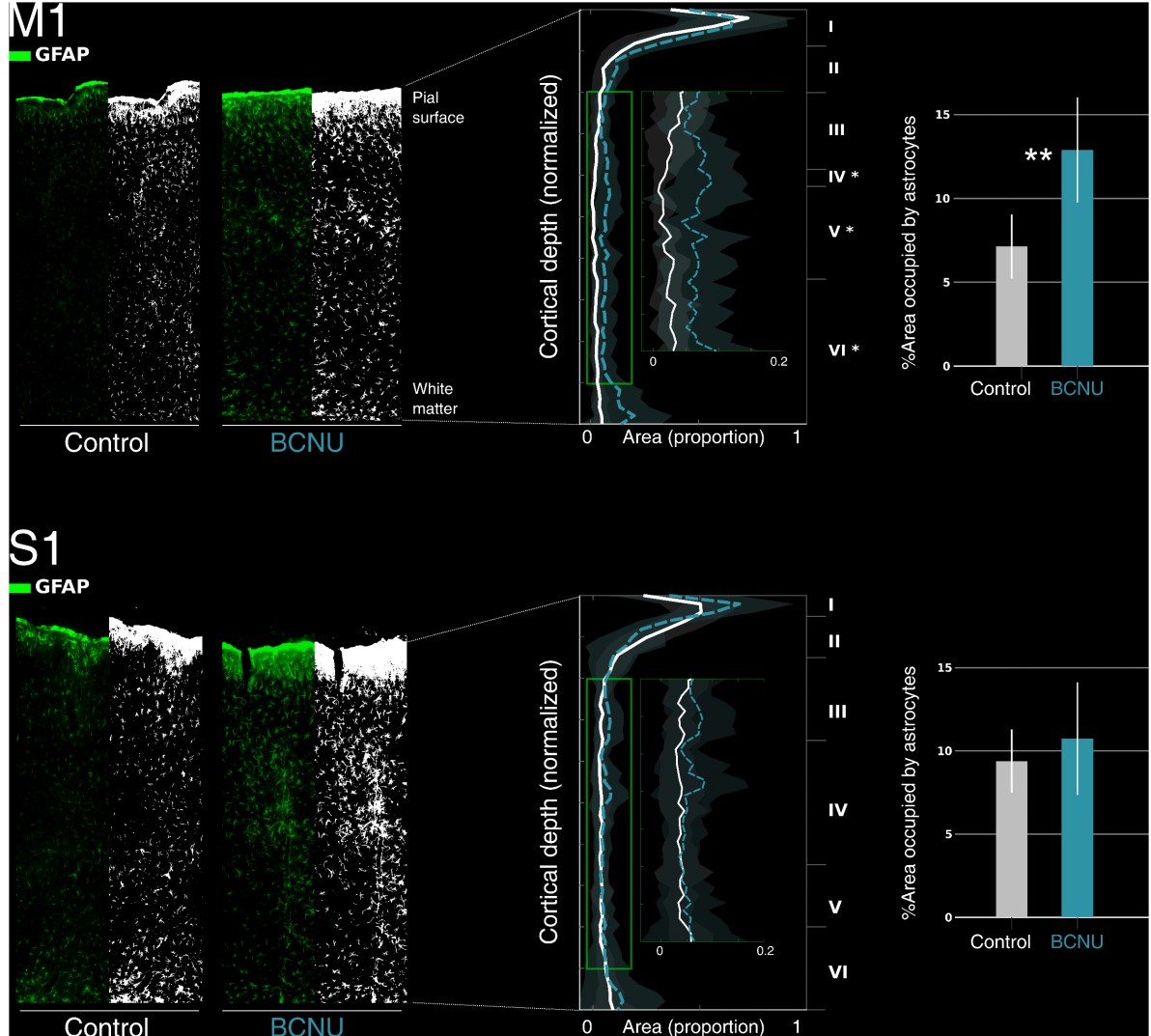

**Fig 6. Quantification of area occupied by astrocytes (GFAP+) in the cortical regions M1 (top row) and S1 (hindlimb region, bottom row).** For each cortical region, left-most panels show the GFAP immunofluorescence (green) in the control and BCNU groups, and the corresponding binarized image (black and white). Middle panels show the group-wise, depth-dependent spatial density profiles, with layers II-V enlarged in the insets. GFAP+ area is larger in experimental animals, particularly in layers IV-VI. Bar graphs in the rightmost panels show between group differences throughout the entire cortical region (**: p < 0.01).

insufficient to characterize the complex organization of the neocortex. Previous reports have shown the utility of diffusion metrics derived from neurite orientation dispersion and density imaging (NODDI) [60], such as microscopic anisotropy and intracellular and intra-neurite volume fractions, for the detection of FCD [57,61,62]. Notably, NODDI is specifically designed to characterize white matter, and therefore may be unfit for the study of the neocortex. Extending the ideas of NODDI, soma and neurite density imaging (SANDI) provides an approximation to the biophysical properties of gray matter with the inclusion of a sphere compartment to model cell bodies [63]. Metrics derived from SANDI are good indicators of gray matter microstructure in animal models [64], as well as in humans [65–67], and therefore constitute a viable option for the detection of FCD in future studies.

The single diffusion encoding method [20] is extremely useful for the acquisition of DWI and their subsequent analyses through many different methods. Nonetheless, acquisition schemes with free gradient waveforms sample the signal more richly across measurement space [29]. From these, complementary diffusion metrics tailored to separate microscopic anisotropy from orientation dispersion and other sources of variance can be obtained through the examination of diffusion tensor distributions [28,29,46]. Acquiring independent observations in a multidimensional manner enhances the characterization of heterogeneous media [30]. In our work, QTI+ metrics showed reductions of microscopic anisotropy that were more extensive than the corresponding DTI abnormalities, as evidenced by the two correlated metrics µFA and $K_{shear}$. These reductions can be interpreted as compromised axonal structure [68–70]. In parallel, $K_{bulk}$ showed an overall increase in the superficial layers of the cortex in BCNU animals, concordant with gliosis and heterogeneity of cellular size and morphology [71,72]. Our histological examinations, as well as prior evaluations of glial cells in BCNU-treated animals [73] support this interpretation. The utility of b-tensor encoding for the detection of different forms of malformations of cortical development has been explored by Lampinen *et al*. [74], with findings in FCD that are in line with our results. Said authors showed relatively low values of microscopic diffusion anisotropy in FCD lesions. Interestingly, regions of increased microscopic anisotropy were observed in the deepest portion of FCD lesions that correspond to the blurred gray/white matter boundary—which the authors interpret as myelin abnormalities of the superficial white matter. In rodents, the cortex is immediately adjacent to very large white matter fascicles (i.e., the corpus callosum and external capsules), contrasting with the relatively large volume of superficial white matter observed in humans [75]. This disposition of fibers makes analysis of juxtacortical white matter of our experimental animals difficult, but in humans the superficial white matter may provide information useful for the identification of cortical malformations [58]. The added value of b-tensor encoding and QTI+, together with the feasibility to perform this type of acquisition efficiently in the clinic [76] open new avenues for the detection of FCD.

The animal model used here shows histopathological features similar to those observed clinically [38], including cortical dyslamination and disarray of the myeloarchitecture [27,35,37]. Reductions in microscopic orientation coherence (CC) were observed in the middle cortex of BCNU-treated animals (Fig 3) and are likely associated with the altered geometric disposition of myelinated fibers (Figs 4 and 5), despite not reaching cluster-level significance. Of note, an increase in glial processes was observed in BCNU-treated animals compared to controls, especially in layers IV-VI of M1 region (Figs 4 and 6). Contrastingly, S1 showed no significant astrocytic density change. Differences of cortical architecture may render M1 more susceptible to microstructural disruption [77,78]. Prominent gliosis due to oxidative stress and inflammation is a common finding in human dysplastic cortex and in many non-genetic animal models [79–81]. While ongoing seizures are a cause of reactive gliosis, astrocytic gliosis alone is capable of initiating epileptic activity [82]. Identification of focal gliosis is, therefore, another opportunity for the detection of FCD, and has in fact been explored with, for example, radiotracers [83] and manganese-enhanced MRI [84]. In agreement with other reports that used dMRI to detect gliosis [85–87], our QTI+ results are indicative of this possibility.

Our study has limitations to consider. While the animal model used here induces alterations of cortical microarchitecture typical of FCD, it does not encompass the range of abnormalities seen in humans. Notably, the cortex of BCNU-treated animals lacks balloon cells, a prominent feature of FCD Type IIb; consequently, the model used more closely resembles FCD Type IIa [88]. Therefore, we cannot ascertain the impact that such abnormal cells may have on the diffusion signal, nor if milder alterations of cyto- and myeloarchitecture characteristic of FCD Type I are enough to alter diffusion metrics. Evaluation of the sensitivity and specificity of diffusion metrics to detect FCD would require a gold standard to be compared to. However, the cortical abnormalities induced by BCNU are not focal but rather distributed across the cortex, which contrasts to the moderately well-demarcated FCD lesions observed in humans. Other animal models, such as freeze lesions of the cortex, induce focal alterations [89], but can include macroscopic abnormalities and the creation of a microgyrus [90]. Such gross anatomical alterations are easily identified with conventional MRI, thus rendering dMRI unnecessary, and are against our ultimate goal to detect the subtle mesoscopic abnormalities that often go unnoticed

in patients. While all animals were processed for histology, technical difficulties during tissue handling preclude a one-to-one comparison of dMRI and microphotographs, and the evaluation of correlations between diffusion metrics and quantitative histology. Diffusion time dependence and water exchange effects, which may modulate QTI+ and other dMRI metrics [91], were not accounted for in this study. This effect may be even more relevant in gray matter given its abundance of non-myelinated (and therefore more water-permeable) axons and dendrites [92–94]. In this study we took the common approach for tuning the waveforms by taking one axis of the STE waveform to construct the LTE waveform, and the other two perpendicular axes to construct the PTE waveform. However, recent studies have shown that the metrics could be biased in the presence of time-dependence effects [95,96]. Future studies should further optimize their acquisition [95] or fitting procedures [96] to properly account for these effects. We attempted to optimize the diffusion gradient waveforms using the NOW toolbox [40] while aiming to minimize echo time as much as possible, which led to waveforms that are somewhat suboptimal. While this is unlikely to impact on the presented results, we warn readers that the diffusion gradient waveforms we present in S1 Fig may be further optimized. Finally, although b-tensor encoding is feasible in clinical scanners [76], spatial resolution will be an important hurdle to overcome in order to provide sufficient sampling of the human cortical mantle, which typically has a thickness of 1.5–4.5 mm [97], thus making super-resolution techniques highly desirable [98].

Our imaging and histology findings paint a coherent picture that highlights the ability of b-tensor encoding to capture the subtle histopathological abnormalities present in FCD. QTI+ resolved fine-grained attributes (microstructural anisotropy loss, orientation dispersion, and intra-voxel heterogeneity) closely tracking myelin disorganization and glial changes. The benefits of b-tensor encoding and QTI+ are especially valuable in the cortex, where the complex architecture of the tissue poses challenges to simpler diffusion models.

## Supporting information

**S1 Fig. Diffusion gradient waveforms.** Three exemplary waveforms are shown for spherical, planar, and linear tensor encodings (STE, PTE and LTE, respectively). STE waveforms were used as a base to create PTE waveforms from its Gx and Gy components, and LTE waveforms using Gz. Gradient amplitude (mT/m) was scaled to create different b value scalings. The table at the top right shows the number of directions for each b-tensor shape and b value scaling. Further optimization of these gradient waveforms is possible and should be considered in future studies.
(TIF)

**S2 Fig. Group comparisons of texture analyses of immunofluorescence microphotographs of myelin basic protein (MBP) in primary motor cortex (M1) and primary somatosensory cortex (hindlimbs region, S1HL).** Depth-wise averages of coherency, energy and radiality (calculated as the absolute dot product between the main orientation of MBP and the normal vector of the pial surface) are shown as mean (lines) ± 1 standard deviation (shared areas). Insets show boxplots for the average values across the entire region, with p-values for individual t-tests.
(TIF)

## Acknowledgments

The authors thank Dr. Juan Ortiz-Retana for assistance during MRI acquisition; Drs. José Martín García Servín, Alejandra Castilla León and María A. Carbajo Mata, for their help at the animal facility. We thank the personnel at the histology facility, particularly Nydia Hernández-Ríos, María Lourdes Palma-Tirado, and Ericka Alejandra De los Ríos; Drs. Remy Fernand Avila Foucat and Reinher Pimentel-Domínguez provided further assistance for immunofluorescence. Image processing was partially performed using the National Laboratory for Advanced Scientific Visualization (LAVIS) with help from Luis Aguilar and Alejandro de León. Additional computing assistance was provided by Leopoldo González-Santos.

## Author contributions

**Conceptualization:** Ricardo Ríos-Carrillo, Hiram Luna-Munguía, Alonso Ramírez-Manzanares, Luis Concha.

**Data curation:** Olimpia Ortega-Fimbres, Mirelta Regalado, Hiram Luna-Munguía, Luis Concha.

**Formal analysis:** Olimpia Ortega-Fimbres, Edith Gaspar-Martínez, Priscila Ruiz-Acosta, Hiram Luna-Munguía, Luis Concha.

**Funding acquisition:** Hiram Luna-Munguía, Luis Concha.

**Investigation:** Olimpia Ortega-Fimbres, Ricardo Ríos-Carrillo, Edith Gaspar-Martínez, Priscila Ruiz-Acosta, Hiram Luna-Munguía, Alonso Ramírez-Manzanares, Luis Concha.

**Methodology:** Olimpia Ortega-Fimbres, Ricardo Ríos-Carrillo, Edith Gaspar-Martínez, Priscila Ruiz-Acosta, Mirelta Regalado, Hiram Luna-Munguía, Alonso Ramírez-Manzanares, Luis Concha.

**Project administration:** Hiram Luna-Munguía, Luis Concha.

**Resources:** Ricardo Ríos-Carrillo, Luis Concha.

**Software:** Ricardo Ríos-Carrillo, Luis Concha.

**Supervision:** Ricardo Ríos-Carrillo, Hiram Luna-Munguía, Alonso Ramírez-Manzanares, Luis Concha.

**Validation:** Hiram Luna-Munguía, Alonso Ramírez-Manzanares, Luis Concha.

**Visualization:** Olimpia Ortega-Fimbres, Luis Concha.

**Writing – original draft:** Olimpia Ortega-Fimbres, Luis Concha.

**Writing – review & editing:** Olimpia Ortega-Fimbres, Ricardo Ríos-Carrillo, Edith Gaspar-Martínez, Priscila Ruiz-Acosta, Mirelta Regalado, Hiram Luna-Munguía, Alonso Ramírez-Manzanares, Luis Concha.

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
