## [Decision Letter · Decision Letter 0]

10 Feb 2026

Dear Dr. Concha,

We look forward to receiving your revised manuscript.

Kind regards,

Jochen Leupold

Academic Editor

PLOS One

Journal Requirements:

https://journals.plos.org/plosone/s/file?id=ba62/PLOSOne_formatting_sample_title_authors_affiliations.pdf....

“Conahcyt/Secihti (CF-2023-I-218 LC) and UNAM-DGAPA (IN204720 and IN213423 LC, IN211326 HLM). The first author received a scholarship from SECIHTI (1019538).”

3. Please note that your Data Availability Statement is currently missing [the repository name and/or the DOI/accession number of each dataset OR a direct link to access each database]. If your manuscript is accepted for publication, you will be asked to provide these details on a very short timeline. We therefore suggest that you provide this information now, though we will not hold up the peer review process if you are unable.

4. Please be informed that funding information should not appear in the Acknowledgments section or other areas of your manuscript. We will only publish funding information present in the Funding Statement section of the online submission form. Please remove any funding-related text from the manuscript.

5. We notice that your supplementary figures are included in the manuscript file. Please remove them and upload them with the file type 'Supporting Information'. Please ensure that each Supporting Information file has a legend listed in the manuscript after the references list.

Additional Editor Comments (if provided):

Dear authors,

first, please apologize the delay in processing your manuscript. The delay was caused by a reviewer who originally agreed to review but never submitted a report.

The final two reviewers are very positive about your manuscript, reviewer 1 suggests to already accept the current version. Reviewer 2 has a number of rather minor comments, so please conisder them in the next version.

Sincerely,

Jochen Leupold

Reviewers' comments:

Reviewer's Responses to Questions

**Comments to the Author**

1. Is the manuscript technically sound, and do the data support the conclusions?

Reviewer #1: Yes

Reviewer #2: Yes

2. Has the statistical analysis been performed appropriately and rigorously?

Reviewer #1: Yes

Reviewer #2: Yes

3. Have the authors made all data underlying the findings in their manuscript fully available?

Reviewer #1: Yes

Reviewer #2: No

4. Is the manuscript presented in an intelligible fashion and written in standard English?

Reviewer #1: Yes

Reviewer #2: Yes

Reviewer #1: The authors have presented interesting data on using MRI to detect focal cortical dysplasia using diffusion tensor imaging. I think this work describes unique efforts to characterized FCDs using MRI and should be interesting to workers in the field.

Reviewer #2: The authors aim to investigate cortical dysplasia in a rat model by diffusion MRI with tensor-valued diffusion encoding and QTI analysis. The investigation focuses on the cortex and results from QTI were compared across control and BCNU-treated animals. Furthermore, an elaborate histological analysis was performed, but it appears that, despite being from the same individuals, the results from MRI and histology were/could not be paired.

I will start by congratulating the authors for performing a difficult and impressive study; it is carefully described and was enjoyable to read. I only have a few relatively minor comments, and I am sure that this study will be appreciated by the community.

Major:

1. The gradient waveform used appears to be a failed optimization attempt. This can be seen from the rapid double-transition across the zero line in several of the plots in the supplementary. I am certain that this does not impact the overall study conclusions, but it should be mentioned as a caution, since readers/authors may want to refrain from using this particular waveform in future studies.

2. I suggest that the authors introduce/explain the parameters uFA, Ki and Ka before the results to aid in the interpretation of figures. Naturally, for QTI experts this is not a problem, but for a wider dMRI audience this could improve the understanding (I like the description in the discussion, but it may come too late). By doing so, the authors may also establish their expectations to how the parameters should relate to histology. Indeed, there are studies that have linked similar analyses to histology, albeit not in FCD.

3. It is disheartening to see that the relation between MRI and histology could not be analysed. But it begs the question how results/conclusions that rely on the connection are affected. I do not mean to impose additional analysis on the authors but rather motivate them disclose this at an earlier stage (in methods) and comment on it more thoroughly in the limitations/discussion. This is important to support/explain statements like Line 246: “advanced dMRI with b-tensor encoding is able to extract information at the mesoscopic level, evidencing subtle histopathological landmarks characteristic of cortical malformations” since this statement appears to rely on an explicit link between histology and MRI. Alternatively, if the problem was identification rather than loss of samples, is it possible to compare analogue parameters across groups? For example, uFA and structure tensor anisotropy for treated vs untreated groups. The data would not be paired, but the group comparisons should show the same direction of the effects.

Minor:

1. Line 58: The generalization of the b-value is mentioned but is slightly unclear. The main addition used in QTI is the shape of the b-tensor, so the transition is rather from a vector- to a tensor-valued entity (not just the scalar b). Also, note that the size of B is the b-value, so the enumeration of features is somewhat overlapping.

2. Line 61+: I enjoy this intro, but it slightly overstates the possibilities of the mentioned methods. For example, it is stated that DTI produces the average D-tensor. This is true, but it is also true for QTI. What QTI adds is a metric of the covariance of “domain D-tensors”. Similarly, it is stated that the DTD can be recovered. This is not quite true. What is possible is to perform an inverse Laplace transform and get a distribution that yields the right signal, but it is unlikely to be accurate. Thus, it is often simplified down to a few descriptives that look like the parameters from QTI. Although I appreciate the positive tone, I would urge the authors to rephrase these statements or perhaps specify the capabilities by referencing specific papers.

3. 2.2: Was the waveform compensated for concomitant gradient effects? If so, please note this.

4. 2.2: It is stated that LTE and PTE were derived from the STE with reference to similar spectral characteristics. I understand that this is a common approach, but it has some limitations. Please consider specifying how the LTE and PTE were selected in more detail, and consider stating that this may still lead to bias, as was recently shown in the following: https://www.biorxiv.org/content/10.64898/2025.12.08.692924v1 (this ref may also support the discussion on diffusion time effects).

5. Line 121: Consider rephrasing “novel” for the parameters from QTI—seeing as they are more than a decade old. Furthermore, since the names Ki and Ka don’t make an appearance in Westin 2016 (correct me if I’m wrong), you may want to add a reference to the paper where these names were introduced, to not confuse a curious reader who is looking for them.

6. Fig 2 and 3: These are very nice figures that do the job well. However, the vertex- and cluster-wise significances are hard to see. The gray circles can be of any size, and in a small figure with gray lines it becomes very challenging to see which regions are below the threshold. I do not have a simple fix, but please consider this for the final version. Furthermore, it seems that the gray circles take on a different color to reflect the effect size, so it is not only gray circles that display significance.

7. Line 289: The similarity to the Lampinen study is mentioned in passing. As they are indeed among the few that use b-tensors in cortical malformations, I think this should be compared more explicitly. E.g., what was similar about the studies, were the conditions, conclusions, or parameters the same/different?

8. Fig 5: The right side of the figure depicts the local orientation of tissue. However, is it not more interesting to show the local anisotropy across some relevant scale? If we are to link this to disorder, I think so. Consider that even when very little structure exists, there will be some orientation reported. Alternatively, the authors could carefully explain that it is the orientational coherence on a larger scale that manifests in parameters like FA.

9. Line 254: Where can I find support from the statement “FA in the middle cortical layers, in close agreement with our previous report”. Is this a visual inspection? If so, please clarify this.

.

Reviewer #1: No

Reviewer #2: No

---

## [Author Response · Author response to Decision Letter 1]

3 Mar 2026

Reviewer #1:

The authors have presented interesting data on using MRI to detect focal cortical dysplasia using diffusion tensor imaging. I think this work describes unique efforts to characterized FCDs using MRI and should be interesting to workers in the field.

We are grateful for the positive reception of our work.

Reviewer #2:

The authors aim to investigate cortical dysplasia in a rat model by diffusion MRI with tensor-valued diffusion encoding and QTI analysis. The investigation focuses on the cortex and results from QTI were compared across control and BCNU-treated animals. Furthermore, an elaborate histological analysis was performed, but it appears that, despite being from the same individuals, the results from MRI and histology were/could not be paired.

I will start by congratulating the authors for performing a difficult and impressive study; it is carefully described and was enjoyable to read. I only have a few relatively minor comments, and I am sure that this study will be appreciated by the community.

Thank you for your kind comments.

Major:

1. The gradient waveform used appears to be a failed optimization attempt. This can be seen from the rapid double-transition across the zero line in several of the plots in the supplementary. I am certain that this does not impact the overall study conclusions, but it should be mentioned as a caution, since readers/authors may want to refrain from using this particular waveform in future studies.

We agree with the Reviewer. Our aim was to minimize TE in the acquisition, and this led to gradient waveforms that were not sufficiently optimized. We have added the following in the limitations of the study:

We attempted to optimize the diffusion gradient waveforms using the NOW toolbox [40] while aiming to minimize echo time as much as possible, which led to waveforms that are somewhat suboptimal. While this is unlikely to impact on the presented results, we warn readers that the diffusion gradient waveforms we present in Figure S1 may be further optimized.

And also in the caption for Figure S1:

Further optimization of these gradient waveforms is possible and should be considered in future studies.

2. I suggest that the authors introduce/explain the parameters uFA, Ki and Ka before the results to aid in the interpretation of figures. Naturally, for QTI experts this is not a problem, but for a wider dMRI audience this could improve the understanding (I like the description in the discussion, but it may come too late). By doing so, the authors may also establish their expectations to how the parameters should relate to histology. Indeed, there are studies that have linked similar analyses to histology, albeit not in FCD.

Thank you for this suggestion, which is also related to your Minor Comment 5 regarding the different nomenclature between Westin (2016) and our work. We expanded the DWI processing paragraph, providing detail regarding the relation of QTI+ metrics to histology. In line with minor comment 5, we now use the nomenclature presented by Westin (2016) throughout the manuscript, thereby changing Ki to Kbulk and Ka to Kshear. We also provide a reference to Nasser et al. (2022), who in their Table 3 provide the relation between different diffusion metrics under various names. The modified text reads:

In addition to the four diffusion-tensor metrics, namely fractional anisotropy (FA), axial, radial and mean diffusivities (AD, RD and MD, respectively) (Fig. 1B, top panel), QTI+ provides four complementary metrics: microscopic anisotropy (µFA), microscopic orientation coherence (CC), and anisotropic and isotropic kurtosis (Kshear, and Kbulk, respectively) [29] (Fig. 1B, bottom panel). µFA quantifies the average anisotropy of compartments (cells) in a voxel disregarding their orientation, while CC is sensitive to the intra-voxel variability of their orientations. Kbulk informs of the heterogeneity in isotropic diffusivities (i.e., tensor sizes) across compartments, and is related to the variance of cell density, edema, and free water. Kshear informs of the variance in microscopic anisotropy (i.e., tensor shapes) across compartments, and helps separate tissue components by their geometry (e.g., axonal from non-axonal components). There is disparity in the literature related to the notation of some diffusion metrics [33]. In this work we use the notation described in [29].

3. It is disheartening to see that the relation between MRI and histology could not be analysed. But it begs the question how results/conclusions that rely on the connection are affected. I do not mean to impose additional analysis on the authors but rather motivate them disclose this at an earlier stage (in methods) and comment on it more thoroughly in the limitations/discussion. This is important to support/explain statements like Line 246: “advanced dMRI with b-tensor encoding is able to extract information at the mesoscopic level, evidencing subtle histopathological landmarks characteristic of cortical malformations” since this statement appears to rely on an explicit link between histology and MRI. Alternatively, if the problem was identification rather than loss of samples, is it possible to compare analogue parameters across groups? For example, uFA and structure tensor anisotropy for treated vs untreated groups. The data would not be paired, but the group comparisons should show the same direction of the effects.

We share the sentiment with the Reviewer. Our plan was to spatially correlate diffusion and quantitative histology metrics. Unfortunately, the laterality of the histological samples was lost during tissue processing, leaving us with no robust way to know left from right hemispheres. We were, however, able to perform between-group comparisons for the GFAP stains (Figure 6). Following the Reviewer’s accurate recommendation, we have performed group comparisons for the structure tensor analyses of the MBP fluorescent stains. This is noted in the Methods (Histological Analysis), and the results shown as Figure S2, which indicates lower coherency of MBP in region M1 (but not in S1HL), and increased energy in region S1HL This new Supplementary Figure serves as an extension of the (now revised) Figure 5.

Minor:

1. Line 58: The generalization of the b-value is mentioned but is slightly unclear. The main addition used in QTI is the shape of the b-tensor, so the transition is rather from a vector- to a tensor-valued entity (not just the scalar b). Also, note that the size of B is the b-value, so the enumeration of features is somewhat overlapping.

We now clarify the generalization of the b-value to the b-tensor. The new text now reads:

In conventional single diffusion encoding, diffusion weighing is applied using gradients of various orientations and magnitudes (b-values), and the encoding can be fully described by a single vector. In MDE, this concept is generalized: the diffusion-encoding vector is replaced by a second-order tensor (the b-tensor), allowing control not only over orientation and magnitude, but also over the shape of the diffusion encoding. Different b-tensor shapes (e.g. linear, planar, or spherical) query the tissue in complementary ways and yield information not reachable by single diffusion encoding. Indeed, while diffusion tensor imaging (DTI) provides characteristics of a single diffusion tensor per voxel, b-tensor encoding adds a metric of the covariance of domain D-tensors, which allows sampling properties of intra-voxel diffusion tensor distribution and provides rich information that better accounts for the heterogeneity of nervous tissue components [32]. These innovations make b-tensor encoded dMRI a promising tool for characterizing the microarchitecture of the cortex and alterations present in FCDs and other cortical malformations [33].

2. Line 61+: I enjoy this intro, but it slightly overstates the possibilities of the mentioned methods. For example, it is stated that DTI produces the average D-tensor. This is true, but it is also true for QTI. What QTI adds is a metric of the covariance of “domain D-tensors”. Similarly, it is stated that the DTD can be recovered. This is not quite true. What is possible is to perform an inverse Laplace transform and get a distribution that yields the right signal, but it is unlikely to be accurate. Thus, it is often simplified down to a few descriptives that look like the parameters from QTI. Although I appreciate the positive tone, I would urge the authors to rephrase these statements or perhaps specify the capabilities by referencing specific papers.

We have modified these sentences, which are part of the modified paragraph indicated in the response to the previous comment.

3. 2.2: Was the waveform compensated for concomitant gradient effects? If so, please note this.

Yes, we used the NOW toolbox to do this. It is now noted in the Methods.

4. 2.2: It is stated that LTE and PTE were derived from the STE with reference to similar spectral characteristics. I understand that this is a common approach, but it has some limitations. Please consider specifying how the LTE and PTE were selected in more detail, and consider stating that this may still lead to bias, as was recently shown in the following: https://www.biorxiv.org/content/10.64898/2025.12.08.692924v1 (this ref may also support the discussion on diffusion time effects).

Thank you for bringing our attention to this potential bias. We have further clarified how the PTE and LTE gradient waveforms are constructed from the tuned STE in the Methods. We now include references to Lasič et al., 2025, and Szczepankiewicz et al, 2025.

The implemented protocol consisted of three b-tensor shapes: linear, spherical and planar [39]. The spherical tensor encoding (STE) gradients were numerically optimized and compensated for concomitant gradients prior to acquisition by using the NOW toolbox (https://github.com/jsjol/NOW) [40, 41] tailored to minimize TE and scaled in magnitude to obtain four b-values (200, 700, 1400, and 2000 s/mm2). To retain gradient spectral characteristics between waveforms, the planar and linear tensor encoding gradients (LTE and PTE, respectively) were extracted from the STE waveform, using one axis for LTE and the other two for PTE [30]. The STE waveform was rotated in 10 directions at each b-value; LTE and PTE waveforms were rotated and scaled to obtain [10, 10, 16, 46] directions for each corresponding b-value.

We added this potential confound in the Discussion.

In this study we took the common approach for tuning the waveforms by taking one axis of the STE waveform to construct the LTE waveform, and the other two perpendicular axes to construct the PTE waveform. However, recent studies have shown that the metrics could be biased in the presence of time-dependence effects [95, 96]. Future studies should further optimize their acquisition [95] or fitting procedures [96] to properly account for these effects.

5. Line 121: Consider rephrasing “novel” for the parameters from QTI—seeing as they are more than a decade old. Furthermore, since the names Ki and Ka don’t make an appearance in Westin 2016 (correct me if I’m wrong), you may want to add a reference to the paper where these names were introduced, to not confuse a curious reader who is looking for them.

We have removed the word “novel” when referring to the acquisition scheme as well as the derived diffusion metrics. As mentioned in the response to the second major point, we now use the notation presented by Westin et al. (2016), substituting Ki to Kbulk and Ka to Kshear throughout the manuscript.

6. Fig 2 and 3: These are very nice figures that do the job well. However, the vertex- and cluster-wise significances are hard to see. The gray circles can be of any size, and in a small figure with gray lines it becomes very challenging to see which regions are below the threshold. I do not have a simple fix, but please consider this for the final version. Furthermore, it seems that the gray circles take on a different color to reflect the effect size, so it is not only gray circles that display significance.

Thank you for this observation. We agree that the gray lines for uncorrected p<0.01 may be too thin to be properly visualized. To maintain the ability to see between-group differences that may not have reached statistical significance, we emphasize Cohen’s d values per data point by keeping a constant marker size (the previous version had marker sizes as |Cohen’s d|). We feel that the updated Figures 2 and 3 are easier to interpret.

7. Line 289: The similarity to the Lampinen study is mentioned in passing. As they are indeed among the few that use b-tensors in cortical malformations, I think this should be compared more explicitly. E.g., what was similar about the studies, were the conditions, conclusions, or parameters the same/different?

The work by Lampinen et al. provides a beautiful demonstration of the utility of tensor-valued dMRI to detect cortical malformations in a clinical setting. Although said article does not include between-group statistical comparisons, and quantitative measures of diffusion metrics are not provided, some direct comparisons between the studies can be made. In particular, FCD lesions showed low, “cortex-like” MKa that is similar to our findings. However, Lampinen et al were able to analyze the juxtacortical white matter and found pockets of high (“white-matter-like”) MKa in the regions that correspond to gray/white matter blurring seen on anatomical images. The anatomical disposition of fibers in rodents precludes the investigation of juxtacortical white matter in our study and we therefore cannot replicate such finding. We have added more details in the Discussion comparing our work to that by Lampinen et al.

The utility of b-tensor encoding for the detection of different forms of malformations of cortical development has been explored [75], with findings in FCD that are in line with our results. Said authors showed relatively low values of microscopic diffusion anisotropy in FCD lesions. Interestingly, regions of increased microscopic anisotropy were observed in the deepest portion of FCD lesions that correspond to the blurred gray/white matter boundary—which the authors interpret as myelin abnormalities of the superficial white matter. In rodents, the cortex is immediately adjacent to very large white matter fascicles (i.e., the corpus callosum and external capsules), contrasting with the relatively large volume of superficial white matter observed in humans [76]. This disposition of fibers makes analysis of juxtacortical white matter of our experimental animals difficult, but in humans the superficial white matter may provide information useful for the identification of cortical malformations [59]. The added value of b-tensor encoding and QTI+, together with the feasibility to perform this type of acquisition efficiently in the clinic [77] open new avenues for the detection of FCD.

8. Fig 5: The right side of the figure depicts the local orientation of tissue. However, is it not more interesting to show the local anisotropy across some relevant scale? If we are to link this to disorder, I think so. Consider that even when very little structure exists, there will be some orientation reported. Alternatively, the authors could carefully explain that it is the orientational coherence on a larger scale that manifests in parameters like FA.

We have modified Figure 5 to better illustrate the lower coherency and disorganization of the MBP+ fibers in the experimental animal as compared to the control animal. We thank the Reviewer for this suggestion.

9. Line 254: Where can I find support from the statement “FA in the middle cortical layers, in close agreement with our previous report”. Is this a visual inspection? If so, please clarify this.

We refer to Villaseñor et al., 2023. While it was cited in the same paragraph, we further clarify it in this new version. Both studies show statistically-significant between-group differences in the same anatomical region (Figure 2B in our current work, and Figure 3A in Villaseñor 2023).

---

## [Decision Letter · Decision Letter 1]

5 Mar 2026

Dear Dr. Concha,

We look forward to receiving your revised manuscript.

Kind regards,

Jochen Leupold

Academic Editor

PLOS One

Journal Requirements:

Additional Editor Comments (if provided):

Dear authors,

please find enclosed a very few minor comments from reviewer 2. (Note the revised version did not go out to reviewer 1 who has already suggested to accept to original submission.)

These remaining comments are so minor that it is not necessary to send the next version out again to reviewer 2. However, to have a more sound processing of the manuscript I think it is better to go for a (quick) minor revision instead of correcting that only in the proof.

Thank you and kind regards,

Jochen Leupold

Reviewers' comments:

Reviewer's Responses to Questions

**Comments to the Author**

Reviewer #2: (No Response)

2. Is the manuscript technically sound, and do the data support the conclusions?

Reviewer #2: Yes

3. Has the statistical analysis been performed appropriately and rigorously?

Reviewer #2: Yes

4. Have the authors made all data underlying the findings in their manuscript fully available?

Reviewer #2: Yes

5. Is the manuscript presented in an intelligible fashion and written in standard English?

Reviewer #2: Yes

Reviewer #2: I am very satisfied with the review and recomend the paper for publication.

I have a few comments that the authors can consider, if they wish, without me needing to see the response.

1. Please check that the translation from Ki and Ka to Kbulk and Kshear is correct. For example, Ka is not equal to Kshear so there should be some factor included that may require extra attention in case this was not done. So please check if you actually intended to use Kµ? I have not visited those calculations in some time, so take this with a grain of salt, but from a quick look at the papers and code I thought it worth warning the authors.

2. The description of Kshear is somewhat off ("Kshear informs of the variance in microscopic anisotropy"). It suggests that a voxel that is described by many identical domain diffusion tensors (all are equally anisotropic but have different orientation such that the variance in anisotropies is zero) should have a Kshear that is zero. But this is not what Kshear or Kµ measure. Please check/rephrase.

Congratulations on your nice work and best wishes,

/FSz

.

Reviewer #2: **Yes:** Filip SzczepankiewiczFilip SzczepankiewiczFilip SzczepankiewiczFilip Szczepankiewicz

---

## [Author Response · Author response to Decision Letter 2]

10 Mar 2026

Reviewer 2

I am very satisfied with the review and recommend the paper for publication. I have a few comments that the authors can consider, if they wish, without me needing to see the response.

We thank Reviewer 2 for the opportunity to clarify the following two details.

1. Please check that the translation from Ki and Ka to Kbulk and Kshear is correct. For example, Ka is not equal to Kshear so there should be some factor included that may require extra attention in case this was not done. So please check if you actually intended to use Kµ? I have not visited those calculations in some time, so take this with a grain of salt, but from a quick look at the papers and code I thought it worth warning the authors.

The metrics provided by QTI+ are Kshear and Kbulk, which is what we report in this work. In the first version of the manuscript we had changed their names to Ka and Ki to (in our mind) facilitate their interpretation. We now see that this adds to the already complicated collection of names for similar diffusion metrics and have decided to retain Kshear and Kbulk.

In addition, acknowledging that Kμ and Kshear are adjacent but not identical metrics, we checked our data and, as expected, found them to have a very high correlation (r=0.996), making the addition of Kμ redundant.

2. The description of Kshear is somewhat off ("Kshear informs of the variance in microscopic anisotropy"). It suggests that a voxel that is described by many identical domain diffusion tensors (all are equally anisotropic but have different orientation such that the variance in anisotropies is zero) should have a Kshear that is zero. But this is not what Kshear or Kµ measure. Please check/rephrase.

We agree that our previous interpretation lacked the contribution of orientational variance. This is now included, with the phrase reading:

Kshear informs of the variance in microscopic anisotropy and dispersion of microenvironments (i.e., tensor shapes and orientations), and helps separate tissue components by their geometry (e.g., axonal from non-axonal components).

---

## [Editor Report · Decision Letter 2]

16 Mar 2026

Analysis of cortical dysplasias using b-tensor encoding diffusion MRI in an animal model

PONE-D-25-64250R2

Dear Dr. Concha,

We’re pleased to inform you that your manuscript has been judged scientifically suitable for publication and will be formally accepted for publication once it meets all outstanding technical requirements.

Kind regards,

Jochen Leupold

Academic Editor

PLOS One

Additional Editor Comments (optional):

Dear Authors,

thank you for the latest corrections, the manuscript is now accepted for publication. Congratulations!

Sincerely,

Jochen Leupold
---

## [Editor Report · Acceptance letter]

PONE-D-25-64250R2

PLOS One

Dear Dr. Concha,

I'm pleased to inform you that your manuscript has been deemed suitable for publication in PLOS One. Congratulations! Your manuscript is now being handed over to our production team.

Kind regards,

on behalf of

Dr. Jochen Leupold

Academic Editor

PLOS One